# Impact of genetic background as a risk factor for atherosclerotic cardiovascular disease: A protocol for a nationwide genetic case-control (CV-GENES) study in Brazil

**Haliton Alves de Oliveira, Junior**[1]*, **Precil Diego Miranda de Menezes Neves**[2], **Gustavo Bernardes de Figueiredo Oliveira**[2], **Frederico Rafael Moreira**[1], **Maria Carolina Tostes Pintão**[3], **Viviane Zorzanelli Rocha**[3], **Cristiane de Souza Rocha**[3], **Viviane Nakano Katz**[3], **Elisa Napolitano Ferreira**[3], **Diana Rojas-Málaga**[3], **Celso Ferraz Viana**[3], **Fabiula Fagundes da Silva**[2], **Juliete Jorge Vidotti**[2], **Natalia Mariana Felicio**[2], **Leticia de Araújo Vitor**[1], **Karina Gimenez Cesar**[1], **Camila Araújo da Silva**[1], **Lucas Bassolli de Oliveira Alves**[1], **Álvaro Avezum**[2]

1 Sustainability and Social Responsibility, Hospital Alemão Oswaldo Cruz, São Paulo, São Paulo, Brazil,
2 International Research Center, Hospital Alemão Oswaldo Cruz, São Paulo, São Paulo, Brazil,
3 Department of Research and Development, Fleury Group, São Paulo, São Paulo, Brazil

\* haliton.oliveira@haoc.com.br

## Abstract

Atherosclerotic Cardiovascular Disease (ASCVD) represents the leading cause of death worldwide, and individual screening should be based on behavioral, metabolic, and genetic profile derived from data collected in large population-based studies. Due to the polygenic nature of ASCVD, we aimed to assess the association of genomics with ASCVD risk and its impact on the occurrence of acute myocardial infarction, stroke, or peripheral artery thrombotic-ischemic events at population level. CardioVascular Genes (CV-GENES) is a nationwide, multicenter, 1:1 case-control study of 3,734 patients in Brazil. Inclusion criterion for cases is the first occurrence of one of the ASCVD events. Individuals without known ASCVD will be eligible as controls. A core lab will perform the genetic analyses through low-pass whole genome sequencing and whole exome sequencing. In order to estimate the independent association between genetic polymorphisms and ASCVD, a polygenic risk score (PRS) will be built through a hybrid approach including effect size of each Single Nucleotide Polymorphism (SNP), number of effect alleles observed, sample ploidy, total number of SNPs included in the PRS, and number of non-missing SNPs in the sample. In addition, the presence of pathogenic or likely pathogenic variants will be screened in 8 genes (*ABCG5*, *ABCG8*, *APOB*, *APOE*, *LDLR*, *LDLRAP1*, *LIPA*, *PCSK9*) associated with atherosclerosis. Multiple logistic regression will be applied to estimate adjusted odds ratios (OR) and 95% confidence intervals (CI), and population attributable risks will be calculated.

**Clinical trial registration:** This study is registered in clinicaltrials.gov (NCT05515653).

**Data Availability Statement:** The repository database containing the genomic data as well as the results of genetic analysis and other relevant clinical, behavioral, and demographic variables will be reported and published after study completion. All relevant data regarding ethics approval, study registration, and original protocol versions are included with the paper.

**Funding:** This study is funded by the Brazilian Ministry of Health, through Adjustment Term 04/2020 – PROADI-SUS. The funder has no role in study design; collection, management, analysis, and interpretation of data; or writing of the report. The national coordination has the final decision to submit any report for publication.

**Competing interests:** The authors have declared that no competing interests exist.

## Introduction

Atherosclerotic cardiovascular disease (ASCVD) represents the leading cause of death worldwide. Overall, one third of deaths is due to cardiovascular disease [1, 2]. Assessment of consistency or variability in the associations between risk factors and cardiovascular disease (CVD) and mortality, both globally and in countries grouped by income levels, will help the development of global and local strategies for prevention. International cohort findings indicate that over 70% of CVD can be attributed to a small cluster of modifiable risk factors [3].

Low-and middle-income countries are substantially affected by CVD. In Brazil, these conditions are responsible for more than 300,000 deaths/year, representing the main cause of death, followed by cancer, respiratory diseases, and diabetes. Collectively, chronic non-communicable diseases (NCDs) are responsible for approximately 70% of the causes of death in both sexes [1, 2, 4, 5].

Contemporary studies have shown that seven out of ten cases of CVD can be explained by modifiable risk factors such as hypertension, low level of education, smoking, dyslipidemia, unhealthy diet, abdominal obesity, sedentary lifestyle, diabetes, psychosocial factors, and air pollution [3]. Furthermore, advances in genetic sequencing technologies and greater access to them have led to the identification of genetic polymorphisms associated with increased risk of CVD, emphasizing the model of interaction between genetic background (non-modifiable) and environment (modifiable) as one of the triggers for CVD [6, 7].

The Brazilian population is unique due to its ethnic diversity, which may be relevant in the assessment of genomics impact on the risk of specific diseases, including CVD [8, 9]. The clinical relevance of genomics in the Brazilian population has been reported in infectious, autoimmune, and hematological diseases, as well as in drug pharmacokinetics and in transplant allocation system.[7, 10–15].

From a public health system perspective, epidemiological features of cardiovascular health in developing countries have been described [16, 17], but there is still an urgent need for studies with geographic representation to allow both regional and ethnic diversity as essential elements to data acquisition and analysis of CVD risk. We designed the case-control study CardioVascular-Genes (CV-GENES) to determine the population attributable fraction estimates of genetic profile in the occurrence of acute myocardial infarction, stroke, or peripheral artery thrombotic-ischemic events, after adjustment for potentially modifiable CVD risk factors in Brazil [18, 19].

## Materials and methods

### Protocol report

CV-GENES protocol version 2.0/2022 was reported through an adaptation of the *Standard Protocol Items: Recommendations for Interventional Trials* (SPIRIT). Fig 1 shows the study schedule of participant enrollment and assessments. The SPIRIT checklist is reported in S1 File. Protocol in original language and its English-translated version is in S2 File.

### Study design

A multicenter observational case-control study will be conducted at 50 centers from all major geographic regions in the country to account for ethnical, cultural, and socioeconomic aspects.

| | Study Period | | |
|---|---|---|---|
| | **Enrolment** | **Allocation** | **Close-out** |
| **TIMEPOINT**** | **$-t_1$** | **0** | **$T_0$** |
| **ENROLMENT:** | | | |
| **Eligibility screen** | X | | |
| **Informed consent** | X | | |
| *Biological samples collection* | | X | |
| **Allocation** | | X | |
| **ASSESSMENTS:** | | | |
| *Clinical, demographic, and behavioral variables* | | X | X |
| *Monogenic hypercholesterolemia* | | X | X |
| *AsCVD polygenic risk score* | | X | X |

**Fig 1. SPIRIT study schedule for participant enrollment and assessments.**

## Eligibility criteria

Cases: All participants aged ≥18 years who are hospitalized due to the first occurrence of an acute myocardial infarction (AMI), stroke, or peripheral artery thrombotic event, presenting in the first 7 days from symptom onset will be screened (S3 File).

Eligible cases must have signs and/or symptoms compatible with the acute manifestation of ASCVD. Case definitions include diagnostic criteria based on national and international guidelines. Acute myocardial infarction is defined as: clinical presentation, electrocardiogram patterns (new pathological Q waves, and/or ST-segment elevation of 1 mm in ≥ 2 contiguous leads, or 2 mm from V1 to V3, or new left bundle branch block, or new changes in the ST-T segment, and cTnI or cTnT curve compatible with acute myocardial injury of ischemic nature (which is currently named a type 1 acute MI). Stroke is defined as clinical signs of rapid development of focal or global brain function disorder, lasting more than 24 hours, with no apparent cause other than that of vascular origin (does not include trauma, cancer, infection of the central nervous system), and Computed Tomography or Magnetic Ressonance Imaging with positive results. Peripheral artery thrombotic event is defined as clinical presentation (pain, burning sensation, absence of pulse, decreased local temperature, changes in color) and diagnostic tests (Doppler ultrasound, angiotomography, or arteriography) that show interruption of flow to the lower limbs acutely, of embolic or thrombotic cause, leading to distal ischemia [20–23].

Controls: All participants aged ≥18 years seeking medical care for non-ASCVD conditions will be screened and enrolled during hospitalization; and healthy individuals with no clinical complain will also be screened and enrolled at the same community. Cases and controls will be recruited in a 1:1 ratio adjusted for sex and age.

Exclusion criteria for both cases and controls are previously known ASCVD, e.g., chronic coronary artery disease, stable angina, previous myocardial infarction, percutaneous revascularization procedures (coronary, cerebrovascular, or peripheral), surgical revascularization procedures in any arterial vascular bed, transient ischemic attack, stroke, aortic aneurysm, intermittent claudication, peripheral artery disease that led to amputation.

## Sample size calculation

We applied the following statistical assumptions: odds ratio for genetics was set at 1.3, allele frequency of 10%, inheritance model was dominant (since the presence of only one polymorphic allele is required to the risk calculation), the prevalence of CVD in the general population of 10%, statistical power of 90%, type I error rate (α) of 0.05, 2-sided significance tests, resulting in a sample of 1,867 cases and 1,867 controls. Sample size calculation was performed using Quanto Software Version 1.2.4.

## Variables

The exposure to modifiable risk factors in combination with genomic data (polygenic risk score) in both cases and controls will be expressed as odds ratios (95% CI). Multiple logistic regression models will be built to adjust and determine the strength of association between demographic variables (sex, age, race), traditional risk factors (smoking, diabetes, hypertension, obesity, anxiety and depression, unhealthy diet, physical inactivity, alcohol consumption, and apolipoprotein B/A1 ratio [ApoB/ApoA1], and genetic data [PRS]. For each variable with a higher chance of CVD (significant OR), attributable risk estimate will be calculated for the genetic component (PRS) and other covariables. Fig 2 shows the Polygenic Risk Score flowchart.

**Fig 2. Polygenic risk score flowchart.**

Table 1 displays the biochemical tests and respective stability and methodology. All biochemical and genetic tests will be run at the same centralized core lab.

### Data source and measurement

Data will be collected through electronic case report forms (eCRFs) specifically designed for this study by using the REDCap software [24]. Demographic variables, prior and inpatient use of medications, biochemical tests (glycosilated hemoglobin, serum apolipoprotein A1, serum apolipoprotein B, total serum cholesterol and fractions [HDL-c, LDL-c, and VLDL-c], serum triglycerides, serum sodium, serum potassium, and measurement of sodium, potassium, creatinine, and albumin in a spot urine sample), and the genetic data (PRS) will be collected.

For the preparation of the whole exome sequencing and low pass whole genome sequencing, the paired-ends libraries will be prepared using 50 ng of DNA as input and library preparation enzymatic fragmentation (EF) Kit 2.0 (Twist Bioscience) with combinatorial dual indexes (Twist Bioscience) reagent kit, following the manufacturer's instructions. Library preparation for WES will include a hybridization capture step with Twist Exome 2.0 probes (Twist Bioscience), following the manufacturer's instructions. Exome and low pass genome will be sequenced on a NovaSeq 6000 platform (IlIumina) using S4 flow cell with 300 cycles (2

**Table 1. Biochemical tests and respective stability and methodology.**

| Test description | Refrigerated Stability | Frozen Stability | Methodology |
|---|---|---|---|
| Glycosilated hemoglobin, whole blood | (2–8 ˚C): 7 days; | (-20 ˚C): 30 days; | Ion exchange Column chromatography in HPLC system |
| Apolipoprotein A-1, serum | (2–8 ˚C): 8 days; | (-20 ˚C): 2 months; | Immunoturbidimetric |
| Apolipoprotein B, serum | (2–8 ˚C): 8 days; | (-20 ˚C): 2 months; | Immunoturbidimetric |
| Cholesterol, serum | (2–8 ˚C): 7 days; | (-20 ˚C): 3 months; | Colorimetric enzyme |
| HDL Cholesterol, serum | (2–8 ˚C): 7 days; | (-20 ˚C): 3 months; | Colorimetric enzyme |
| Cholesterol, LDL Fraction, serum | (2–8 ˚C): 7 days; | (-20 ˚C): 3 months; | Calculation based on Friedewald formula |
| Cholesterol, VLDL Fraction, serum | (2–8 ˚C): 7 days; | (-20 ˚C): 3 months; | Martin and colleagues' formula |
| Triglycerides, serum | (2–8 ˚C): 7 days; | (-20 ˚C): 1 year; | Colorimetric enzyme |
| Creatinine, serum | (2–8 ˚C): 5 days; | (-20 ˚C): 1 year. | Colorimetric kinetic |
| Sodium, serum | (2–8 ˚C): 7 days; | (-20 ˚C): 6 months; | Potentiometric |
| Potassium, serum | (2–8 ˚C): 7 days; | (-20 ˚C): 6 months; | Potentiometric |
| Sodium, spot, urine | (2–8 ˚C): 7 days; | (-20 ˚C): 6 months; | Potentiometric |
| Potassium, urine | (2–8 ˚C): 7 days; | (-20 ˚C): 3 months; | Potentiometric |
| Creatinine, spot, urine | (2–8 ˚C): 5 days; | (-20 ˚C): 1 year; | Colorimetric kinetic |
| Albuminuria, spot, urine | (2–8 ˚C): 14 days; | (-20 ˚C): 6 months; | Immunoturbidimetric assay |

x 150 bp—paired end). Run data and quality control will be monitored in NovaSeq control software. For the preparation of the whole exome sequencing and low pass whole genome sequencing, the paired-ends libraries will be prepared using 50 ng of DNA as inputs and enzymatic fragmentation and combinatorial dual indexes (Twist Bioscience) reagent kit, following the manufacturer's instructions. Exome and low pass genome will be sequenced on a NovaSeq 6000 platform (IlIumina) using S4 flow cell with 300 cycles (2 x 150 bp—paired end). Run data and quality control will be monitored in NovaSeq control software.

The genetic evaluation will be performed through the association of low-pass whole genome sequencing (WGS) (coverage 0.5-1x) with data imputation and whole exome sequencing (WES) (average coverage 90x). The method of PRS calculation is described in the S3 File [25–27]. In addition, the presence of pathogenic or likely pathogenic variants will be screened in 8 genes (*ABCG5*, *ABCG8*, *APOB*, *APOE*, *LDLR*, *LDLRAP1*, *LIPA*, *PCSK9*) associated with atherosclerosis, from the WES (S3 File) [28–30]. For patients in whom novel monogenic variants are identified, those variants will be classified for their pathogenicity according to The American College of Medical Genetics and Genomics (ACMG) criteria [31]. For the Polygenic Risk Score calculation, only Single Nucleotide Polymorphisms previously associated with an increased cardiovascular risk will be selected.

Both protocols (WGS and WES) were validated previously to this study by using standard control and samples with known results (orthogonal tests) and the data were compared. In addition, coverage, Polymerase Chain Reaction duplicates and other parameters were evaluated.

## Data analysis

**Quantitative variables and statistical analysis.** The Hardy-Weinberg genetic equilibrium test will be evaluated in the control group using the chi-square test or Fisher's exact test. To estimate the association between genetic polymorphisms and risk of CVD, univariate and multiple unconditional logistic regression analyses will be conducted [32–36].

Initially, univariate binary logistic regression analyses will be performed. Thus, covariates with a p-value < 0.20 in univariate regression analyses will be considered in multiple logistic

regression analysis with selection of variables according to the backward elimination technique [37, 38]. The selection process in the backward elimination allows the sample size to adjust and always uses the maximum available sample as the number of variables in the model drops [37]. Occasionally, covariates judged as confounding factors by the investigator may be forced into the final backward regression model.

Additional multiple logistic regression analyses may also be conducted using a variant of the Purposeful Selection algorithm described by Bursac *et al.* [38].

The assumption of linearity on the logit scale (log-odds) between each quantitative covariate and the binary response variable in binary logistic regression analysis will be evaluated with the construction of "Smoothed Scatter Plots" and the method of fractional polynomials [35–39]. When the assumption is not satisfied, quantitative covariates will be categorized for use in logistic regression using cut-off points, according to the literature, distribution tertiles or optimal cut-off obtained from the Receiver Operating Characteristic (ROC) curve, whichever is deemed most appropriate. In case of an optimal cut-off point, it will be defined as the one that maximizes the Youden index [39, 40].

**Estimation of population attributable risk.** To characterize the study population, categorical variables will be described as counts and proportions and compared with Pearson's chi-square test or with Fisher's exact test [41]. Normally and asymmetrically distributed quantitative variables will be expressed as mean (standard deviation) or median (interquartile range), respectively [42]. Normality will be assessed by visually inspecting histograms and applying normality tests, if appropriate [43, 44]. Comparison of these continuous type variables will be carried out with Student's t test for independent samples or Mann-Whitney test for non-parametric distribution [41].

All statistical analyses will follow the complete case analysis principle. All hypothesis tests will be 2-sided and p-value<0.05 considered statistically significant. Statistical data analysis will be conducted with SAS 9.4 (SAS Institute, Cary, NC).

**Bias control.** Eligibility criteria for cases and controls will be strictly followed to reduce selection bias. The criteria defined for the first ASCVD event were standardized, and all sites have been trained for the correct identification, data recording, and reporting of information. Cases will be validated by the adjudication committee through clinical data, diagnostic tests, hospitalization and/or discharge summary, among others, using the standardized and validated process for diagnoses [45, 46]. The relationship between the variables will be adjusted by multivariate logistic models. Data monitoring will be applied throughout the entire study timeline to assure data quality. Blood and urine samples collection have been standardized, and genetic and biochemistry analyses will be performed in a core lab to guarantee the standardization of methods and quality control.

## Study organization

International Research Center at Hospital Alemão Oswaldo Cruz is responsible for the national coordination, clinical monitoring, data quality assurance, and adjudication of cases and controls. All biochemistry tests and genetic tests es analyses will be conducted by Fleury Group S.A.

## National and local approvals & ethics in the analysis

This study has been approved by the Institutional Review Board (IRB) at Hospital Alemão Oswaldo Cruz and by all local Ethics Committees at each participating site (CAAE—56482922.2.1001.0070), on April 22nd, 2022. Any changes will be treated as protocol amendments and will be immediately submitted to IRB.

All participants who agree to be enrolled must sign two informed consent forms (one agreement with the study participation and another agreement with the biological samples storage into the national coordination biobank). A specific anonymized 'id' will be assigned to each participant and their biological samples. Data will be stored in a secure and encrypted cloud.

This study is registered in clinicaltrials.gov (NCT05515653) and is currently recruiting. First patient was recruited in July 2022. The estimated completion date is April 2024.

## Discussion

We have designed and launched a nationwide multicenter observational case-control study to determine the population attributable fraction estimate of genetics in the overall risk of first occurrence of ASCVD. We will be able to assess the polygenic risk score and ASCVD and also a set of specific monogenic inheritance genes associated with atherosclerosis at the population level. This national effort will provide clinical and genetic data to support the National Program in Genomics and Precision Medicine (GENOMAS BR), which currently aims to screen 100,000 Brazilians.

AMI worldwide is associated with nine modifiable risk factors (ApoB/ApoA1 ratio, hypertension, diabetes, abdominal obesity, smoking, psychosocial factors, poor diet, regular alcohol consumption and lack of regular physical activity). Beyond these nine above-mentioned risk factors, one additional factor, cardiac risk, is also associated with stroke. These factors account for 90% of the population attributable risk (PAR) of both acute myocardial infarction and stroke. Therefore, it is biologically plausible that the genetic background might also play a role in the remaining 10% of PAR of ASCVD.

In this study, the contribution of the genetic background in the risk of ASCVD among Brazilian individuals will be ascertained by the analysis of variants with a potential direct effect of disease manifestation (monogenic diseases). The polymorphisms will be ascertained in genes that compound the risk in association to each other (polygenic diseases), which behave as factors of susceptibility to a disease, in the "two hits" pathogenic model, where there is a direct interaction of a genetic background with environmental factors for the development of the disease [8–10].

A previous study of 8,795 individuals of European, South Asian, Arab, Iranian, and Nepalese origin from the INTERHEART case-control study performed a genotyping of 1,536 SNPs from 103 genes evaluated as well as analyzed previous established cardiovascular risk factors. Thirteen polymorphisms were associated with increased cardiovascular risk, 11 of which were related to serum levels of APOB/A1 ratio, 1 associated to LDL-cholesterol receptor and 1 to Apolipoprotein E [7].

The INTERHEART study showed that genetic risk markers for the occurrence of AMI, identified through genome analysis, appear to be broadly associated with AMI in several different ethnic groups. However, the population attributable risk to these genetic factors is relatively small compared to that of modifiable risk factors. In the Latin American subgroup, the impact of the genetic risk score was PAR 0.86 95% CI 0.74–0.93 and PAR 0.12 95% CI 0.02–0.42, respectively, for modifiable and non-modifiable factors. However, as already mentioned, there are no data exclusively from Brazilian individuals with adequate statistical power and standardized data collection of traditional risk factors and, therefore, recommended for an integrated assessment of the PRS impact on the PAR for the occurrence of atherosclerotic events.

In a review of the main studies of susceptibility loci for coronary artery disease, including those from important consortia such as CARDIoGRAM, MIGen, WTCCC and Cardiogenics, deCODE, CARDIoGRAM, C4D and CARDIoGRAM + C4D [47], important contributions were derived from the identification of 60 loci, whose mechanism of action is related to serum

levels of LDL-cholesterol, lipoprotein (a) and triglycerides, blood pressure, obesity, coagulation profile, changes in endothelial and smooth muscle cells of the vascular wall, migration mechanisms and cell adhesion, immune activation, inflammation, cell growth, differentiation and apoptosis, as well as extracellular matrix constituents and also some loci with unknown function.

Polymorphisms in genes associated with the regulation of calcium levels (CASR, CYP24A1, CARS, DGKD, DGKH/KIAA0564 and GATA3) [48] and metalloproteinases (MMP-3 and MMP-9) [49] were also associated with an increased risk of coronary artery disease and AMI. The list of polymorphisms associated with AMI progressively increases, and some authors have already reinforced the importance of using the genetic background in cardiovascular risk calculators [50–52].

Regarding stroke cases, polymorphisms in some genes such as MTHFR, eNOS, ACE, AGT, ApoE, PON1, PDE4D were associated with a higher risk of ischemic stroke. For cases of hemorrhagic stroke, polymorphisms in collagen genes, TLR4 and CD14 and even the gene that gives rise to C-reactive protein were identified [53].

Studies have shown that genetic background may exert a significant effect on increased risk of cardiovascular disease [54]. In a mixed population such as ours, determining the attributable risk may lead to the development of actions aimed at specific subpopulations, such as providing appropriate genetic advice and prevention strategies. This is particularly important considering that lifestyle changes and even correct adherence to pharmacological therapies are not implemented or followed in a sustained way.

The contribution of the genetic component on the pattern or susceptibility to diseases in the Brazilian population has already been shown to have an impact on the control and treatment of infectious, autoimmune, and hematological diseases, drug pharmacokinetics, and even on the transplant allocation system [13, 14, 19, 55, 56].

It is worth to cite the potential impact of epigenetics to the pathogenesis of human diseases [57–59]. Currently, the role of control of gene expression by non-coding RNAs is a hot topic of research including cardiovascular diseases [60–62]. Nevertheless, our study aims to assess the association of genomics with ASCVD risk and its impact on the occurrence of acute myocardial infarction, stroke, or peripheral artery thrombotic-ischemic events at population level. Epigenetics studies may be warranted and of interest for future research and study protocols.

The CV-GENES study has some potential limitations. First, since Brazil is a large country with different demographic densities and socioeconomic disparities, assuring equality of individuals allocation based on geographic parameters is challenging. Second, the gold standard genetic test for this purpose is whole genomic sequencing with high cover and depth. Facing the size of our cohort and limited financial support, we decided to combine the whole exome sequencing with low-pass whole genome sequencing to allow for PRS calculation, a successful strategy used by previous studies and supported by guidelines [26, 27, 63, 64].

In this context, the detection of polymorphisms associated with CVD could identify patients in whom modifiable risk factors can be screened/treated early, aiming at cardiovascular prevention. Similarly, these results might help design and implement specific screening programs for early detection and prompt risk factors control to reduce the burden of premature cardiovascular morbidity and mortality among those with higher PRS combined with cluster of modifiable risk factors.

## Supporting information

**S1 File. SPIRIT checklist.**
(DOCX)

**S2 File. CV-GENES research protocol.**
(PDF)

**S3 File. Eligibility criteria and genetic evaluation.**
(DOCX)

## Acknowledgments

This study was conceived with the support of the Brazilian Unified Health System Institutional Development Program (PROADI-SUS). We would like to thank Dr. Antônio José Cordeiro Mattos for his important contributions in the early stage of this protocol.

## Author Contributions

**Conceptualization:** Haliton Alves de Oliveira, Junior, Precil Diego Miranda de Menezes Neves, Gustavo Bernardes de Figueiredo Oliveira, Frederico Rafael Moreira, Maria Carolina Tostes Pintão, Viviane Zorzanelli Rocha, Cristiane de Souza Rocha, Viviane Nakano Katz, Elisa Napolitano Ferreira, Álvaro Avezum.

**Data curation:** Precil Diego Miranda de Menezes Neves, Gustavo Bernardes de Figueiredo Oliveira, Fabiula Fagundes da Silva.

**Formal analysis:** Haliton Alves de Oliveira, Junior, Frederico Rafael Moreira.

**Funding acquisition:** Haliton Alves de Oliveira, Junior.

**Investigation:** Precil Diego Miranda de Menezes Neves, Gustavo Bernardes de Figueiredo Oliveira, Juliete Jorge Vidotti, Álvaro Avezum.

**Methodology:** Haliton Alves de Oliveira, Junior, Precil Diego Miranda de Menezes Neves, Frederico Rafael Moreira, Maria Carolina Tostes Pintão, Viviane Zorzanelli Rocha, Cristiane de Souza Rocha, Viviane Nakano Katz, Elisa Napolitano Ferreira, Diana Rojas-Málaga, Celso Ferraz Viana, Fabiula Fagundes da Silva, Natalia Mariana Felicio, Lucas Bassolli de Oliveira Alves.

**Project administration:** Haliton Alves de Oliveira, Junior, Leticia de Araújo Vitor, Karina Gimenez Cesar, Camila Araújo da Silva.

**Resources:** Maria Carolina Tostes Pintão, Viviane Zorzanelli Rocha, Cristiane de Souza Rocha, Viviane Nakano Katz, Elisa Napolitano Ferreira.

**Software:** Cristiane de Souza Rocha.

**Supervision:** Haliton Alves de Oliveira, Junior, Fabiula Fagundes da Silva, Álvaro Avezum.

**Validation:** Haliton Alves de Oliveira, Junior, Precil Diego Miranda de Menezes Neves, Gustavo Bernardes de Figueiredo Oliveira, Lucas Bassolli de Oliveira Alves, Álvaro Avezum.

**Writing – original draft:** Haliton Alves de Oliveira, Junior, Precil Diego Miranda de Menezes Neves, Gustavo Bernardes de Figueiredo Oliveira, Frederico Rafael Moreira, Maria Carolina Tostes Pintão, Viviane Zorzanelli Rocha, Cristiane de Souza Rocha, Viviane Nakano Katz, Elisa Napolitano Ferreira, Diana Rojas-Málaga, Celso Ferraz Viana, Fabiula Fagundes da Silva, Juliete Jorge Vidotti, Natalia Mariana Felicio, Leticia de Araújo Vitor, Karina Gimenez Cesar, Camila Araújo da Silva, Lucas Bassolli de Oliveira Alves, Álvaro Avezum.

**Writing – review & editing:** Haliton Alves de Oliveira, Junior, Precil Diego Miranda de Menezes Neves, Gustavo Bernardes de Figueiredo Oliveira, Frederico Rafael Moreira,

Maria Carolina Tostes Pintão, Viviane Zorzanelli Rocha, Cristiane de Souza Rocha, Viviane Nakano Katz, Elisa Napolitano Ferreira, Diana Rojas-Málaga, Celso Ferraz Viana, Fabiula Fagundes da Silva, Juliete Jorge Vidotti, Natalia Mariana Felicio, Leticia de Araújo Vitor, Karina Gimenez Cesar, Camila Araújo da Silva, Lucas Bassolli de Oliveira Alves, Álvaro Avezum.

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
