## [Decision Letter · Decision Letter 0]

25 Sep 2023

PONE-D-23-20597Impact of Genetic Background as a Risk Factor for Atherosclerotic Cardiovascular Disease: A Protocol for a Nationwide Genetic Case-Control (CV-GENES) study in BrazilPLOS ONE

Dear Dr. Alves Oliveira Junior,

Thank you for submitting your manuscript to PLOS ONE. After careful consideration, we feel that it has merit but does not fully meet PLOS ONE’s publication criteria as it currently stands. Therefore, we invite you to submit a revised version of the manuscript that addresses the points raised during the review process.

We look forward to receiving your revised manuscript.

Kind regards,

Mohammad Reza Mahmoodi, Ph.D.

Academic Editor

PLOS ONE

4. Please remove your figures from within your manuscript file, leaving only the individual TIFF/EPS image files, uploaded separately. These will be automatically included in the reviewers’ PDF.

Reviewers' comments:

Reviewer's Responses to Questions

**Comments to the Author**

1. Does the manuscript provide a valid rationale for the proposed study, with clearly identified and justified research questions?

Reviewer #1: Yes

Reviewer #2: Yes

Reviewer #3: Yes

2. Is the protocol technically sound and planned in a manner that will lead to a meaningful outcome and allow testing the stated hypotheses?

Reviewer #1: Yes

Reviewer #2: Yes

Reviewer #3: Partly

3. Is the methodology feasible and described in sufficient detail to allow the work to be replicable?

Reviewer #1: Yes

Reviewer #2: Yes

Reviewer #3: No

4. Have the authors described where all data underlying the findings will be made available when the study is complete?

Reviewer #1: Yes

Reviewer #2: No

Reviewer #3: Yes

5. Is the manuscript presented in an intelligible fashion and written in standard English?

Reviewer #1: Yes

Reviewer #2: No

Reviewer #3: No

6. Review Comments to the Author

You may also provide optional suggestions and comments to authors that they might find helpful in planning their study.

Reviewer #1: The authors described a proposed study protocol that will be used to assess the relationship between inherited genetic variants that are potentially associated with the incidence of Atherosclerotic Cardiovascular Disease in the Brazilian population. The protocol is well designed and complies to the ethical standards that are usually applied in such studies. Consequently, I recommend accepting this manuscript for publication.

Reviewer #2: The study protocol entitled “Impact of Genetic Background as a Risk Factor for Atherosclerotic Cardiovascular Disease: A Protocol for a Nationwide Genetic Case-Control (CV-GENES) study in Brazil” through various methods investigates the alterations in human genome associated with cardiovascular disease. The manuscript is well-written and well-designed, and the utilized methods sufficiently address the goals of the design. Moreover, such study has an undeniable importance in precision medicine, genetic counseling, and uncovering complexities in human genome.

Some issues to be address are:

A reference is needed for: “… 8 genes (ABCG5, ABCG8, APOB, APOE, LDLR, LDLRAP1, LIPA, PCSK9) associated with atherosclerosis. “

The study suffers from plagiarism. Authors must paraphrase their sentences.

Does the pipeline of this study protocol follow any previously-published publication? If yes, authors should add references for that.

Did authors include only “previously reported” Pathogenic & Likely Pathogenic variants or they also included novel ones? If both types were included, more details should be mentioned.

Minor Grammar & punctuation changes are required such as: in the first page of the Introduction, the “Low- and middle- income countries“ should be changed to “Low- and middle-income countries”. Or, the “Sample‐ size calculation” should be corrected too.

It would be great if authors can define the full description of CV-GENES if any meaning it has! If it refers to cardiovascular genes, it should be fully mentioned in its first appearance of the term.

Authors should also mention the “Study limitations”.

Also, as another suggestion, authors can include a section as National and local approvals & Ethics in the Analysis to have a more structured demonstration.

Regarding the Brazilian parts in supplementary file, I don’t know Brazilian and someone else should confirm the accuracy of the info and data provided there.

Authors should describe that what confirmation tests they will perform for the novel variant that they find in this study?

One more important thing, authors should explain what steps they have considered for population stratification regarding criteria such as gender, different descents, or etc.

Reviewer #3: Dear authors

Thanks for performing such a great work. The subject is interesting and a necessity for every society. However, there are some issues that is better to be clarified in the manuscript as follows:

1- As we know when reporting a work that has been done in the past we should use past tense for reporting. Unfortunately in the manuscript all verb tenses refer to future; we will perform, will be done etc. which should be corrected throughout the text.

2- There is no mention how the selected (8) genes were chosen for the study.

3- There is no appropriate description of methods, devices and kits and reagents used in the study.

4- There is no proper description of statistical analysis for data obtained with the large cohort of patients.

5- It appears you did WES in the study. Did you do WES for all patients?

6- I did not find percentage of mutations for studied genes. Which gene was more involved in the disease?

7- How did you validated the data obtained by sequencing?

8- Your methodology was based on WES, but your conclusion was based on gene polymorphysim (last paragraph of discussion). Did you investigate polymorphysim of the selected genes? If so please describe the method used for polymorphysim study.

9- We know that many epigenetic alterations from non-coding RNAs (miRNAs, lncRNAs, CircRNAs) affect gene expression. There is no mention of such effects in the discussion that should be considered seriously.

7. PLOS authors have the option to publish the peer review history of their article (what does this mean?). If published, this will include your full peer review and any attached files.

Reviewer #1: **Yes: **Abdallah Ahmad Medlej

Reviewer #2: **Yes: **Maziar Ganji

Reviewer #3: **Yes: **Hossein Mozdarani

---

## [Author Response · Author response to Decision Letter 0]

7 Nov 2023

São Paulo, November 7th, 2023

Dear Editor,

We would like to kindly thank you for your reply and we are very grateful to the editors and the reviewers for the assessment of our manuscript “Impact of Genetic Background as a Risk Factor for Atherosclerotic Cardiovascular Disease: A Protocol for a Nationwide Genetic Case-Control (CV-GENES) study in Brazil” to be considered for publication in PLOS ONE as a Design Paper. Their relevant and helpful comments and suggestions significantly improved our manuscript. 

We confirm that this manuscript has not been published elsewhere and is not under consideration by another journal. All authors have approved the manuscript, agreed with its submission, and have no conflicts of interest to disclose. Responses to the editors and reviewers are outlined below.

JOURNAL REQUIREMENTS:

Response: We verified the PLOS ONE’s style requirements and performed all the corrections. Changes are highlighted in the manuscript. 

Response: The correct grant information was inserted into both ‘Funding Information’ and ‘Financial Disclosure’ sections. 

Upon re-submitting your revised manuscript, please upload your study’s minimal underlying data set as either Supporting Information files or to a stable, public repository and include the relevant URLs, DOIs, or accession numbers within your revised cover letter. For a list of acceptable repositories, please see http://journals.plos.org/plosone/s/data-availability#loc-recommended-repositories. 

Any potentially identifying patient information must be fully anonymized.

Response: The current study integrates the GENOMAS-BRAZIL, a Nationwide program that aims to genotype Brazilian individuals in diverse clinical settings. Currently, the program is still defining the public repository database. The repository database containing our genomic data as well as the final results of genetic analysis and other relevant clinical, behavioral and demographic variables will be made available after the publication of the final results. We would like to emphasize that this is a design/protocol paper, therefore, final results are not available since the study is currently recruiting cases and controls. All relevant information regarding ethics approval, study registration, and original protocol versions were uploaded into this submission. 

We updated the ‘Data availability statement’ section to deal with your suggestion. Please see the text between lines 363-369. 

4. Please remove your figures from within your manuscript file, leaving only the individual TIFF/EPS image files, uploaded separately. These will be automatically included in the reviewers’ PDF.

Response: We removed the figures from the main manuscript as requested.

Response: Thanks for this remark. Supporting information section was inserted at the end of the document. Please, see the text between lines 601-605 

Response: All references were reviewed and there is no retracted reference in the current list. There is an ‘ERRATUM’ available for three references (Ref 3. Yusuf et al., 2020, Ref. 21 Sacco et al., 2013 and Ref. 22 Callum et al., 2000). Nevertheless, online versions used have already been updated. 

REVIEWERS' COMMENTS:

Review Comments to the Author

REVIEWER #1

The authors described a proposed study protocol that will be used to assess the relationship between inherited genetic variants that are potentially associated with the incidence of Atherosclerotic Cardiovascular Disease in the Brazilian population. The protocol is well designed and complies to the ethical standards that are usually applied in such studies. Consequently, I recommend accepting this manuscript for publication.

Response: Thank you for your kind comments and recommendation. 

REVIEWER #2

The study protocol entitled “Impact of Genetic Background as a Risk Factor for Atherosclerotic Cardiovascular Disease: A Protocol for a Nationwide Genetic Case-Control (CV-GENES) study in Brazil” through various methods investigates the alterations in human genome associated with cardiovascular disease. The manuscript is well-written and well-designed, and the utilized methods sufficiently address the goals of the design. Moreover, such study has an undeniable importance in precision medicine, genetic counseling, and uncovering complexities in human genome. Some issues to be address are:

1. A reference is needed for: “… 8 genes (ABCG5, ABCG8, APOB, APOE, LDLR, LDLRAP1, LIPA, PCSK9) associated with atherosclerosis.”

Response: We have inserted the references accordingly.

28. Musunuru K, Hershberger RE, Day SM, Klinedinst NJ, Landstrom AP, Parikh VN, Prakash S, Semsarian C, Sturm AC; American Heart Association Council on Genomic and Precision Medicine; Council on Arteriosclerosis, Thrombosis and Vascular Biology; Council on Cardiovascular and Stroke Nursing; and Council on Clinical Cardiology. Genetic Testing for Inherited Cardiovascular Diseases: A Scientific Statement From the American Heart Association. Circ Genom Precis Med. 2020 Aug;13(4):e000067. doi: 10.1161/HCG.0000000000000067.

29. Watts GF, Gidding SS, Hegele RA, Raal FJ, Sturm AC, Jones LK, Sarkies MN, Al-Rasadi K, Blom DJ, Daccord M, de Ferranti SD, Folco E, Libby P, Mata P, Nawawi HM, Ramaswami U, Ray KK, Stefanutti C, Yamashita S, Pang J, Thompson GR, Santos RD. International Atherosclerosis Society guidance for implementing best practice in the care of familial hypercholesterolaemia. Nat Rev Cardiol. 2023 Jun 15. doi: 10.1038/s41569-023-00892-0.

30. Cuchel M, Raal FJ, Hegele RA, Al-Rasadi K, Arca M, Averna M, Bruckert E, Freiberger T, Gaudet D, Harada-Shiba M, Hudgins LC, Kayikcioglu M, Masana L, Parhofer KG, Roeters van Lennep JE, Santos RD, Stroes ESG, Watts GF, Wiegman A, Stock JK, Tokgözoğlu LS, Catapano AL, Ray KK. 2023 Update on European Atherosclerosis Society Consensus Statement on Homozygous Familial Hypercholesterolaemia: new treatments and clinical guidance. Eur Heart J. 2023 Jul 1;44(25):2277-2291. doi: 10.1093/eurheartj/ehad197.

2. The study suffers from plagiarism. Authors must paraphrase their sentences.

Response: Thank you for your concern, but please allow us to clearly state that this is an original research with its related manuscript. After your comment, we carried out a new plagiarism check using the Turnitin tool. In fact, our manuscript was cited by MedRxiv, a preprint server for distribution of preliminary reports of research manuscripts in health sciences that have not been certified by peer review. This preprint server has gained relevance during the COVID-19 pandemic to provide open and fast access to data being gathered to help clinicians and health professionals dealing with severe cases. We cannot interfere or prohibit them to cite submitted works before peer review. Definitely, this should not be considered as plagiarism. 

3. Does the pipeline of this study protocol follow any previously-published publication? If yes, authors should add references for that.

Response: Our protocol is based on previous internationally published studies with additional correction of genetic attributable risk for well-established risk factors for cardiovascular diseases. We added the references to the main manuscript as suggested. Please the references citation on line 178 and references list in the end of the manuscript:

25. Al-Jumaan M, Chu H, Alsulaiman A, Camp SY, Han S, Gillani R, et al. Interplay of Mendelian and polygenic risk factors in Arab breast cancer patients. Genome Med. 2023 Sep 1;15(1):65. doi: 10.1186/s13073-023-01220-4.

26. O'Sullivan JW, Raghavan S, Marquez-Luna C, Luzum JA, Damrauer SM, Ashley EA, et al; American Heart Association Council on Genomic and Precision Medicine; Council on Clinical Cardiology; Council on Arteriosclerosis, Thrombosis and Vascular Biology; Council on Cardiovascular Radiology and Intervention; Council on Lifestyle and Cardiometabolic Health; and Council on Peripheral Vascular Disease. Polygenic Risk Scores for Cardiovascular Disease: A Scientific Statement From the American Heart Association. Circulation. 2022 Aug 23;146(8):e93-e118. doi: 10.1161/CIR.0000000000001077. 

27. Homburger JR, Neben CL, Mishne G, Zhou AY, Kathiresan S, Khera AV. Low coverage whole genome sequencing enables accurate assessment of common variants and calculation of genome-wide polygenic scores. Genome Med. 2019;11:74. doi: 10.1186/s13073-019-0682-2.

4. Did authors include only “previously reported” Pathogenic & Likely Pathogenic variants or they also included novel ones? If both types were included, more details should be mentioned.

Response: Our first aim is to include only previously reported variants when analyzing monogenic variants. For the PRS calculation we will select only SNPs already reported to be related to cardiovascular diseases as well.

5. Minor Grammar & punctuation changes are required such as: in the first page of the Introduction, the “Low- and middle- income countries“ should be changed to “Low- and middle-income countries”. Or, the “Sample‐ size calculation” should be corrected too.

Response: Thanks for your detailed review. Indeed, some minor typo errors remained after text revision in the track changes format with little spaces between words. We have reviewed the current version and made the required corrections. These changes are marked along the whole manuscript. It will be easier to visualize them in the track changes version of the manuscript. 

6. It would be great if authors can define the full description of CV-GENES if any meaning it has! If it refers to cardiovascular genes, it should be fully mentioned in its first appearance of the term.

Response: We mentioned the acronym on its first citation in the abstract and introduction sections of the manuscript (Pages 2 and 3, respectively). 

7. Authors should also mention the “Study limitations”.

Response: Thank you for your recommendation. We added a paragraph with study limitations on Discussion section, as suggested. Please, see the text inserted between lines 334-341. 

8. Also, as another suggestion, authors can include a section as National and local approvals & Ethics in the Analysis to have a more structured demonstration.

Response: Thank you for your suggestion. We highlighted this section on Methods, as suggested. Information regarding IRB approval, ethical requirements and study registration are depicted in this section. Please, see the text between lines 241-253. 

9. Regarding the Brazilian parts in supplementary file, I don’t know Brazilian and someone else should confirm the accuracy of the info and data provided there.

Response: The documents refer to the protocol that were approved by Ethics Committee in our native language (Portuguese). This document was attached as a requirement from PLOS ONE Editor. Nevertheless, both Portuguese and English versions of the protocol are attached as supplementary files. Furthermore, the protocol has been registered in clinicaltrials.gov. 

10. Authors should describe that what confirmation tests they will perform for the novel variant that they find in this study?

Response: Thank you for your remark. For patients in whom novel monogenic variants are identified, those variants will be classified for their pathogenicity according to ACMG criteria. For the Polygenic Risk Score calculation, only Single Nucleotide Polymorphisms previously associated with an increased cardiovascular risk will be selected. Please, see the text inserted between lines 180-185. We have also inserted a reference for ACMG criteria: 

31. Richards S, Aziz N, Bale S, Bick D, Das S, Gastier-Foster J, et al. ACMG Laboratory Quality Assurance Committee. Standards and guidelines for the interpretation of sequence variants: a joint consensus recommendation of the American College of Medical Genetics and Genomics and the Association for Molecular Pathology. Genet Med. 2015 May;17(5):405-24. doi: 10.1038/gim.2015.30.

11. One more important thing, authors should explain what steps they have considered for population stratification regarding criteria such as gender, different descents, or etc.

Response: Please note that, for this study, we have been recruiting cases and controls matching the sex and age (plus or minus 5 years) inclusion criteria as a suggestion of recruitment procedure. Furthermore, we clearly stated that we will perform an unconditional statistical analysis to include the entire dataset. This method should avoid reduction in sample size statistical power and complete dataset will also be adjusted for age and sex. This strategy was adopted, for example, in the INTERHEART and INTERSTROKE studies. Additionally, to enable our public health system representativeness, recruitment will be stratified by type of hospitals, i.e., publics hospitals will account for 70% of the entire set of institutions. Furthermore, controls who are relatives of cases included will not be recruited in order to avoid genetic inheritance bias between groups. 

REVIEWER #3

Dear authors, 

Thanks for performing such a great work. The subject is interesting and a necessity for every society. However, there are some issues that is better to be clarified in the manuscript as follows:

1- As we know when reporting a work that has been done in the past we should use past tense for reporting. Unfortunately in the manuscript all verb tenses refer to future; we will perform, will be done etc. which should be corrected throughout the text.

Response: Thank you for your comment. Please note that the current manuscript represents a “Study Protocol paper” (journals.plos.org) that aims to report the protocol rationale, background, objectives, design, and methods of an ongoing study, which has not been finished yet. This justifies the use of future tenses. 

2- There is no mention how the selected (8) genes were chosen for the study.

Response: Those 8 genes were selected according to recommendations of international guidelines of lipid disorders and inherited cardiovascular diseases. We added this information to the manuscript and inserted the references. Please, see the text between lines 178-180. References: 

28 Musunuru K, Hershberger RE, Day SM, Klinedinst NJ, Landstrom AP, Parikh VN, et al. American Heart Association Council on Genomic and Precision Medicine; Council on Arteriosclerosis, Thrombosis and Vascular Biology; Council on Cardiovascular and Stroke Nursing; and Council on Clinical Cardiology. Genetic Testing for Inherited Cardiovascular Diseases: A Scientific Statement From the American Heart Association. Circ Genom Precis Med. 2020 Aug;13(4):e000067. doi: 10.1161/HCG.0000000000000067.

29. Watts GF, Gidding SS, Hegele RA, Raal FJ, Sturm AC, Jones LK, et al. International Atherosclerosis Society guidance for implementing best practice in the care of familial hypercholesterolaemia. Nat Rev Cardiol. 2023 Jun 15. doi: 10.1038/s41569-023-00892-0.

30. Cuchel M, Raal FJ, Hegele RA, Al-Rasadi K, Arca M, Averna M, Bruckert E, Freiberger T, Gaudet D, Harada-Shiba M, Hudgins LC, et al. 2023 Update on European Atherosclerosis Society Consensus Statement on Homozygous Familial Hypercholesterolaemia: new treatments and clinical guidance. Eur Heart J. 2023 Jul 1;44(25):2277-2291. doi: 10.1093/eurheartj/ehad197.

3- There is no appropriate description of methods, devices and kits and reagents used in the study.

Response: Methods are fully described in Supplementary material 3. We included this additional information to methods section, as follows (Please, see the text between lines 160-174: 

“For the preparation of the whole exome sequencing and low pass whole genome sequencing, the paired-ends libraries will be prepared using 50 ng of DNA as input and library preparation enzymatic fragmentation (EF) Kit 2.0 (Twist Bioscience) with combinatorial dual indexes (Twist Bioscience) reagent kit, following the manufacturer´s 

instructions. Library preparation for WES will include a hybridization capture step with Twist Exome 2.0 probes (Twist Bioscience), following the manufacturer´s instructions. Exome and low pass genome will be sequenced on a NovaSeq 6000 platform (IlIumina) using S4 flow cell with 300 cycles (2 x 150 bp - paired end). Run data and quality control will be monitored in NovaSeq control software. For the preparation of the whole exome sequencing and low pass whole genome sequencing, the paired-ends libraries will be prepared using 50 ng of DNA as inputs and enzymatic fragmentation and combinatorial dual indexes (Twist Bioscience) reagent kit, following the manufacturer´s instructions. Exome and low pass genome will be sequenced on a NovaSeq 6000 platform (IlIumina) using S4 flow cell with 300 cycles (2 x 150 bp - paired end). Run data and quality control will be monitored in NovaSeq control software.”

Furthermore, to better explore the methods in the abstract, we have inserted this text between lines 35-39: “In order to estimate the independent association between genetic polymorphisms and ASCVD, a polygenic risk score (PRS) will be built through a hybrid approach including effect size of each Single Nucleotide Polymorphism (SNP), number of effect alleles observed, sample ploidy, total number of SNPs included in the PRS, and number of non-missing SNPs in the sample”.

4- There is no proper description of statistical analysis for data obtained with the large cohort of patients.

Response: Thank you for the comment. This is a Study Protocol (Design) paper, and we did describe the main statistical analysis plan. Therefore, no data analyses have been done yet since participant recruitment and data collection have not been completed yet. We briefly described the statistical analysis in Methods section. We highlighted the descriptive statistics, the logistic regression adjustment for the OR and the population attributable risk estimate. Specific data regarding calculation of Polygenic Risk Score and clinical data are described in S3 File.

5- It appears you did WES in the study. Did you do WES for all patients?

Response: This is correct. WES will be performed for all subjects of the study (cases and controls), and we will also perform a low pass genome analysis for all participants. 

6- I did not find percentage of mutations for studied genes. Which gene was more involved in the disease?

Response: Indeed, this will be a great point that we will certainly highlight when the study is finalized. This manuscript is a Study Protocol paper, the whole genetic analysis has not been performed yet. Sequencing of 8 genes associated with atherosclerosis (ABCG5, ABCG8, APOB, APOE, LDLR, LDLRAP1, LIPA, PCSK9) will be performed during the study, and data will be analyzed and published in 2024/2025. However, as a short preview to answer your question, currently 2011 DNA samples have been sequenced and only 460 monogenic analyzes were performed, with only five identified mutations (LDLR e APOB). 

7- How did you validated the data obtained by sequencing?

Response: Thanks for this question. Please note that this is a Study Protocol manuscript, and the whole genetic analysis has not been completed yet. Sequencing of low pass genome and exome will be performed during the study. Both protocols (WGS and WES) were validated previously to this study by using standard control and samples with known results (orthogonal tests) and the data were compared. In addition, coverage, PCR duplicates and other parameters were evaluated. We inserted this information in lines 186-189. 

8- Your methodology was based on WES, but your conclusion was based on gene polymorphysim (last paragraph of discussion). Did you investigate polymorphysim of the selected genes? If so please describe the method used for polymorphysim study.

Response: Thank you for your relevant point. Indeed, this will be considered for the final analysis. This is a Study Protocol manuscript, and no results or conclusions are being discussed here. As we mentioned in Methods section of the manuscript: “The genetic evaluation will be performed through the association of low-pass whole genome sequencing (WGS) (coverage 0.5-1x) with data imputation and whole exome sequencing (WES) (average coverage 90x) to obtain PRS score. The method of PRS calculation is described in the S3 File.”

In this context, the detection of polymorphisms associated with cardiovascular disease could identify patients where modifiable risk factors can be screened/treated early, aiming at primary cardiovascular prevention. In the same way, results might help design and implement specific screening aimed at early detection and prompt risk factors control to reduce the burden of premature cardiovascular morbidity and mortality among those with higher PRS combined with cluster of modifiable risk factors. We will certainly discuss all relevant variables and the entire dataset once we complete the statistical analysis. 

9- We know that many epigenetic alterations from non-coding RNAs (miRNAs, lncRNAs, CircRNAs) affect gene expression. There is no mention of such effects in the discussion that should be considered seriously.

Response: This is a very good comment, nonetheless, epigenetic is not our aim in this project. The primary aim of the current study is to assess the association of genomics (Polygenic Risk Score - PRS) with ASCVD risk and its impact on the occurrence of acute myocardial infarction, stroke, or peripheral artery thrombotic-ischemic events at a population level. Since this is a relevant question, we added a paragraph regarding epigenetics on the Discussion section. Please, see the text between lines 327-333. References: 

57. Geiger M, Gorica E, Mohammed SA, Mongelli A, Mengozi A, Delfine V, et al. Epigenetic Network in Immunometabolic Disease. Adv Biol (Weinh). 2023 Oct 4:e2300211. doi: 10.1002/adbi.202300211.

58. Wong H, Sugimura R. Immune-epigenetic crosstalk in haematological malignancies. Front Cell Dev Biol. 2023 Sep 21;11:1233383. doi: 10.3389/fcell.2023.1233383.

59. Mallick R, Duttaroy AK. Epigenetic modification impacting brain functions: Effects of physical activity, micronutrients, caffeine, toxins, and addictive substances. Neurochem Int. 2023 Oct 11;171:105627. doi: 10.1016/j.neuint.2023.105627.

60. McCarthy K, O'Halloran AM, Fallon P, Kenny RA, McCrory C. Metabolic syndrome accelerates epigenetic ageing in older adults: Findings from The Irish Longitudinal Study on Ageing (TILDA). Exp Gerontol. 2023 Oct 24;183:112314. doi: 10.1016/j.exger.2023.112314.

61. Prasher D, Greenway SC, Singh RB. The impact of epigenetics on cardiovascular disease. Biochem Cell Biol. 2020 Feb;98(1):12-22. doi: 10.1139/bcb-2019-0045.

62. Wołowiec A, Wołowiec Ł, Grześk G, Jaśniak A, Osiak J, Husejko J, et al. The Role of Selected Epigenetic Pathways in Cardiovascular Diseases as a Potential Therapeutic Target. Int J Mol Sci. 2023 Sep 6;24(18):13723. doi: 10.3390/ijms241813723.

Yours sincerely,

---

## [Decision Letter · Decision Letter 1]

21 Nov 2023

Impact of Genetic Background as a Risk Factor for Atherosclerotic Cardiovascular Disease: A Protocol for a Nationwide Genetic Case-Control (CV-GENES) study in Brazil

PONE-D-23-20597R1

Dear Dr. Alves Oliveira Junior,

We’re pleased to inform you that your manuscript has been judged scientifically suitable for publication and will be formally accepted for publication once it meets all outstanding technical requirements.

Kind regards,

Mohammad Reza Mahmoodi, Ph.D.

Academic Editor

PLOS ONE

Additional Editor Comments (optional):

Reviewers' comments:

Reviewer's Responses to Questions

**Comments to the Author**

1. Does the manuscript provide a valid rationale for the proposed study, with clearly identified and justified research questions?

Reviewer #2: Yes

2. Is the protocol technically sound and planned in a manner that will lead to a meaningful outcome and allow testing the stated hypotheses?

Reviewer #2: Yes

3. Is the methodology feasible and described in sufficient detail to allow the work to be replicable?

Reviewer #2: Yes

4. Have the authors described where all data underlying the findings will be made available when the study is complete?

Reviewer #2: Yes

5. Is the manuscript presented in an intelligible fashion and written in standard English?

Reviewer #2: Yes

6. Review Comments to the Author

You may also provide optional suggestions and comments to authors that they might find helpful in planning their study.

Reviewer #2: All mentioned issues in the first review were sufficiently addressed. Congratulations for this great work!

7. PLOS authors have the option to publish the peer review history of their article (what does this mean?). If published, this will include your full peer review and any attached files.

Reviewer #2: **Yes: **Maziar Ganji

---

## [Editor Report · Acceptance letter]

3 Mar 2024

PONE-D-23-20597R1 

PLOS ONE

Dear Dr. Alves de Oliveira Junior, 

I'm pleased to inform you that your manuscript has been deemed suitable for publication in PLOS ONE. Congratulations! Your manuscript is now being handed over to our production team.

Kind regards, 

on behalf of

Dr. Mohammad Reza Mahmoodi 

Academic Editor

PLOS ONE